# Path Planning Research of a UAV Base Station Searching for Disaster Victims’ Location Information Based on Deep Reinforcement Learning

**DOI:** 10.3390/e24121767

**Published:** 2022-12-02

**Authors:** Jinduo Zhao, Zhigao Gan, Jiakai Liang, Chao Wang, Keqiang Yue, Wenjun Li, Yilin Li, Ruixue Li

**Affiliations:** Zhejiang Integrated Circuits and Intelligent Hardware Collaborative Innovation Center, Hangzhou Dianzi University, Hangzhou 310018, China

**Keywords:** UAV base station, path planning, deep reinforcement learning, received signal strength

## Abstract

Aiming at the path planning problem of unmanned aerial vehicle (UAV) base stations when performing search tasks, this paper proposes a Double DQN-state splitting Q network (DDQN-SSQN) algorithm that combines state splitting and optimal state to complete the optimal path planning of UAV based on the Deep Reinforcement Learning DDQN algorithm. The method stores multidimensional state information in categories and uses targeted training to obtain optimal path information. The method also references the received signal strength indicator (RSSI) to influence the reward received by the agent, and in this way reduces the decision difficulty of the UAV. In order to simulate the scenarios of UAVs in real work, this paper uses the Open AI Gym simulation platform to construct a mission system model. The simulation results show that the proposed scheme can plan the optimal path faster than other traditional algorithmic schemes and has a greater advantage in the stability and convergence speed of the algorithm.

## 1. Introduction

Nowadays, with the development of 5G communication technology and the improvement of the mobile Internet of Things, unmanned aerial vehicle (UAV) base stations have begun to be widely used in auxiliary communication and post-disaster rescue tasks [1,2]. The UAV air base station has the characteristics of strong maneuverability, controllable mobility, and convenient deployment and can support the high-speed transmission of communication data, etc. [3]. The application of UAV base stations in disaster relief scenarios improves the problem of communication signals being difficult to reach in complex environments, and also makes it possible to provide all-round coverage of communication signals in disaster relief mission areas. Considering the special nature of UAV base stations and the complexity of the disaster relief environment [4,5], how to use the high flexibility of UAVs to plan an optimal collision-free path in the complex situation with obstacles after a disaster is the most pending problem of UAVs in disaster rescue missions.

In the scenario where UAVs search for location information of disaster victims, the path planning problem of UAVs is the key to accomplish such tasks. Traditional UAV path planning algorithms include the artificial potential field method, heuristic algorithm, ant colony algorithm, etc. The UAV path planning method using an improved artificial potential field was proposed in [6,7], which effectively solved the path planning of multi-UAS and UAV obstacle avoidance against dynamic obstacles by introducing a rotating potential field and Markov prediction model. In [8,9,10], an improved heuristic algorithm was proposed to solve the problem of UAV path planning and mission area coverage in complex environments. To solve the coverage path planning problem for autonomous heterogeneous UAVs over a finite area, an original clustering-based algorithm was designed in [11], which divided the mission area into clusters to obtain the optimal UAV point-to-point path. For the multi-target search and path planning problem in unknown environment, Refs [12,13] proposed an improved artificial bee colony algorithm, which greatly improved the stability of UAV flight and the speed of UAV path planning. In [14], a combination of a pseudospectra algorithm and an ant colony algorithm was used to solve the 3D path planning problem of solar powered UAV. In [14], a combination of the pseudospe-tral and colony algorithms was used to solve the 3D path planning problem of solar powered UAV. For the path planning problem of multiple UAVs, the authors of [15,16] proposed the use of a sparrow search algorithm and a two-level coordination framework approach to achieve path planning for UAV swarms in dynamic obstacle environments, respectively. In order to improve the stability of the algorithm and the efficiency of the task completion during the task execution, designing a new alternative framework for UAV motion performance can effectively reduce the task execution time of UAVs while improving the robustness of the algorithmic framework [17,18]. A non-rigid hierarchical discrete grid structure was proposed in [19] to achieve path planning of UAVs in 3D space. These optimization algorithms based on traditional algorithms were all based on converting the UAV path planning problem into a path optimization problem and solving the optimization model to obtain the optimal flight path. However, these methods often suffer from a low intelligence level, a long solution time and restricted application scenarios, and cannot be applied to post-disaster rescue scenarios with complex environments and changing scenarios.

In recent years, artificial intelligence technology has flourished, and deep reinforcement learning has gradually been applied to solve the path planning problems of UAVs in various complex environments. In [20,21], the use of the DDPG algorithm was proposed to enable UAVs to autonomously avoid threat areas and thus obtain an optimal flight path. The improved DRL algorithm was used in [22,23,24] to implement real-time UAV path planning in the respective application scenarios. In [25], to improve the collection of global and local information during the flight of UAVs, a multi-layer path planning algorithm based on reinforcement learning (RL) technique was proposed, which divided the information into upper and lower layers and then coordinated the processing of the upper and lower layers to finally plan a collision-free path for the UAV. By introducing a UAV mobile edge computing platform in [26,27,28], a better quality of path planning for reinforcement learning algorithms was provided while risk avoidance was achieved. In scenarios where there is no basic communication infrastructure or where communication with the Internet is not possible due to emergencies such as disasters, a reinforcement learning-based path planning scheme for IoT UAVs was proposed in [29] to achieve autonomous path planning for UAVs in unknown environments. A DL-based collision avoidance method for UAV communication networks was proposed in [30], while a series of convex optimization problems were formulated and solved to effectively solve the optimal trajectory planning problem for UAV communication networks. To optimize the flight trajectory and improve the energy management capability of the solar powered aircraft, a neural network controller was trained using the RL method in [31] and used as an integrated controller for aircraft navigation and guidance. In [32,33], the algorithms of dual-latency deep deterministic policy gradient (TD3) participant-critic deep reinforcement learning (DRL) framework and multi-step duel DDQN (multi-step D3QN) were used to implement UAV path design in 3D space, respectively. For the path planning problem of multi-UAV wireless data collection, the process was solved by the DRL method in [34] and an approximate optimal UAV control strategy with unknown environmental data information was implemented.

Traditional reinforcement learning algorithms and traditional path planning algorithms often have good test effects in some simple scenarios with simple environments and small state dimensions of algorithm input, but the execution effect of the algorithms will be greatly reduced in some scenarios with complex environments, many obstacles, and large state information dimensions input by the algorithms.

For the shortcomings of traditional path planning algorithms, this paper studies the path planning problem of UAV base stations collecting location information of trapped people based on deep reinforcement learning technology, proposes the reinforcement learning idea of combining state splitting and optimal value, and also designs the DDQN-SSQN reinforcement learning algorithm to solve the path planning problem of UAVs in performing search and rescue tasks by combining the received signal strength indication (RSSI) collected by UAV base stations in real time during the flight. Finally, the DDQN-SSQN algorithm is used to simulate the built task environment and compare with the traditional reinforcement learning algorithm to demonstrate the advantages of the proposed algorithm. The proposed algorithm improves the network structure of the algorithm on the basis of the traditional DDQN algorithm, and the algorithm classifies the multi-dimensional environmental state and conducts targeted training, which greatly improves the decision-making efficiency of the Agent.

The main contributions of this paper are as follows:

(1) The path decision-making algorithm based on deep reinforcement learning is used to classify and store multi-dimensional state information, and the optimal path information is obtained through targeted training, which greatly improves the efficiency of UAV to complete search and rescue tasks;

(2) Use the self-built virtual environment model to complete the training of the drone, which effectively avoids the training loss of the physical UAV and reduces the cost of model training;

(3) Introduce the received signal strength indication (RSSI) into the algorithm model of deep reinforcement learning and use this to affect the reward obtained by the agent, which reduces the difficulty of UAV decision-making and improves the positioning accuracy of the ground user’s location coordinates.

The structure of this article is as follows: In the second chapter, the construction of a virtual disaster relief scenario is introduced. The construction of the scene model is mainly completed from three aspects: system environment model, obstacle model, and ground user model. In the third chapter, the design ideas of the DDQN-SSQN algorithm proposed in this paper and the network structure of the algorithm are introduced. In the fourth chapter, the relevant parameters of the algorithm simulation are given, and the test effect of the proposed algorithm and the traditional reinforcement learning algorithm is compared to verify the advantages of the proposed new algorithm. The full text is summarized and prospected in Section 5.

## 2. Model Construction

This chapter mainly introduces the construction of the UAV search ground disaster personnel model. The model describes the environmental information for the UAV search and rescue mission and also provides a training ground for the UAV. The system model transforms the UAV path planning problem into a reward and punishment decision problem by introducing decision theory and a related mathematical model of the cost function. In order to simulate the disaster relief environment of UAV in real situations, the whole mission model is modeled from three aspects: system environment, obstacles, and ground users.

### 2.1. System Environment Model

Figure 1 shows the system environment model of the UAV base station performing search and rescue missions for trapped people.
$S_{xy}=L_{x}\times L_{y}$ is the area of the entire mission area. The entire scene model is divided into multiple subtask areas according to the number of UAVs, and the area of each subtask area is $S_{xy}^{i}$. N, $N=\left\{1,2,3,\ldots N\right\}$ is the number of drones used to perform search and rescue missions in the scenario, M, $M=\left\{1,2,3,\ldots M\right\}$ is the number of people affected on the ground. The position coordinates of the UAV change with increasing time and can be expressed as:(1)$$P_{o}^{n}\left(t\right)=\left[x_{o}^{n}\left(t\right),y_{o}^{n}\left(t\right),z_{o}^{n}\left(t\right)\right]\in R^{3}$$

In the equation, t denotes the time of flight of the UAV and $t\in \left[0,T^{n}\right],n\in \left[0,N\right],T^{n}$ indicate the time required for the *n* UAVs to complete the search and rescue mission. Considering that the ground users will be trapped in the debris when the house collapses and the movement of the trapped people is often restricted by the spatial environment, the ground trapped people are set to appear randomly at any location of the scene model and the location coordinates do not change with the increase in time. The spatial coordinates of the trapped people can be expressed as: (2)$$P_{i}^{m}=\left[x_{i}^{m},y_{i}^{m},z_{i}^{m}\right]\in R^{3}$$

Here, $m\in \left[0,M\right]$. K, $K=\left\{1,2,3,\ldots K\right\}$ indicates the number of obstacles in the scene. Since the position of obstacles in the mission scene generally does not move after the house collapses, the location of the set obstacle is fixed in the scene. The coordinates of the spatial location of the obstacle are as follows:(3)$$P_{c}^{k}=\left[x_{c}^{k},y_{c}^{k},z_{c}^{k}\right]$$
(4)$$x_{c}^{k},y_{c}^{k}\in R$$
(5)$$z_{c}^{k}\in \left[0,H\right]$$

In the formula, H denotes the height of the obstacle from the horizontal ground.

The entire area to be searched will be divided into N subtask areas according to the performance of N UAVs, and a UAV will be arranged on each subtask area to collect the location information of trapped people, and so as to achieve the coverage of the entire mission area by N UAVs. The area of N subtask regions is $S_{xy}^{i},i=\left\{0,1,2\ldots N\right\}$. The area $S_{xy}$ of the scene model can be expressed as:(6)$$S_{xy}=\overset{N}{\underset{i=0}{\sum }}S_{xy}^{i}$$

The size of the N subtask area $S_{xy}^{i}$ obtained from the scene model segmentation will be set according to the performance of N UAVs. The performance of the drone will be judged based on a combination of indicators such as the drone’s endurance, flight speed, and climb rate. The performance of N UAVs in the scenario is $\left\{I^{1},I^{2},I^{3},\ldots ,I^{N}\right\}$, The area of the task area $S_{xy}^{i}$ obtained by segmentation can be expressed as:(7)$$S_{xy}^{i}=S_{xy}\times \frac{I^{i}}{\sum_{n=1}^{N}I^{n}}$$

Here, $i=\left\{0,1,2\ldots N^{\prime }\right\}$.

### 2.2. Obstacle Model

UAVs inevitably encounter obstacles in the actual flight process. Since there is randomness in whether an obstacle will appear in the UAV’s line-of-sight range, the obstacle model can be built by random generation. The virtual scene dynamically adjusts the number of obstacles K in the environment according to the size of the actual task area and places the obstacles randomly at any location in the scene. Considering the UAV’s own load and the search effect of the UAV, the UAV needs to fly close to the ground when performing rescue missions, which leads to the UAV having to avoid various obstacles such as trees, houses, and ruins during the process of moving. In order to enable the UAV to better detect the surrounding environment, this paper used millimeter-wave radar technology to realize the real-time monitoring and identification of obstacles during the flight of the UAV. The back-end signal processing module of millimeter-wave radar can use the echo signal to calculate the presence, speed, direction, distance, angle and other target information of moving objects. At the same time, millimeter-wave radar also has the characteristics of small size and light weight, so it can be carried around the airframe without affecting the UAV load, and effectively complete the obstacle avoidance task.

The obstacle avoidance detection of the UAV is shown in Figure 2. As can be seen in the figure, the position coordinates of the UAV in flight are $\left(x_{o},y_{o}\right)$, the distance between the UAV and the obstacle is $d_{i}$, and the phase difference is $\beta_{i}$. The two-dimensional coordinates of the obstacle $\left(x_{i}^{\prime },y_{i}^{\prime }\right)$ obtained by the UAV monitoring during the flight can be expressed as:(8)$$\left\{\begin{array}{c} x_{i}^{\prime }=x_{o}+d_{i}\times cos\beta_{i}\\ y_{i}^{\prime }=y_{o}+d_{i}\times sin\beta_{i} \end{array}\right.$$

Here, $i\in \left[0,K\right]$. In the figure, $v_{u}$ denotes the velocity of the UAV flight, $v_{u\_x}$ denotes the component velocity of the UAV along the x-axis direction, and $v_{u\_y}$ denotes the component velocity of the UAV along the y-axis direction. The flight speed of the UAV at the moment t+1 and the component speeds on the x and y axes can be expressed as:(9)$$v_{u}\left(t+1\right)=v_{u}\left(t\right)+ât$$
(10)$$v_{u\_x}\left(t+1\right)=v_{u\_x}\left(t\right)+ât\times \cos\theta$$
(11)$$v_{u\_y}\left(t+1\right)=v_{u\_y}\left(t\right)+ât\times \sin\theta$$

In the formula, â denotes the flight acceleration of the UAV, and θ denotes the angle between the actual flight direction of the UAV and the x-axis.

The UAV will monitor the location information of surrounding obstacles in real time through the millimeter wave radar mounted around the body during flight. The signal processing module of the UAV will transmit a feedback signal $f_{c}$ to the UAV based on the distance $d_{ci}$ between the body and the obstacle, and the signal $f_{c}$ can be expressed as:(12)$$f_{c}=\left\{\begin{array}{c} 1,\;\;\;\;\;\;\;\;\;\;\;\;\;15\leq d_{ci}\\ \;\;\;\;0,\;\;\;\;\;\;\;D_{c}\leq d_{ci}\leq 15\\ \;-1,\;\;\;\;\;\;\;\;0\leq d_{ci}\leq \;D_{c} \end{array}\right.\;$$

Here, $d_{ci}$ denotes the distance between the UAV and the obstacle, and $D_{c}$ denotes the danger radius where the UAV is at risk of collision. Equation (12) can be seen, when the horizontal distance between the UAV and the obstacle is greater than 15m, the threat of the obstacle to the UAV is very small, at this time can be considered that there is no obstacle around the body, that is, f=1; when the distance is greater than the danger radius of the UAV is less than 15m, at this time can be considered that the UAV is approaching the obstacle, that is, f=0; when the distance is less than the danger radius $D_{c}$, the UAV may collide with the obstacle at any time, at this time can be considered mission failure, that is, f=−1.

### 2.3. Ground User Model

This scenario model is used to simulate a situation where people are buried when a building collapses. The UAV carries an airborne base station over the mission area and collects in real time the received signal strength between the cell phones of the affected people on the ground and the base station. The formula for calculating RSSI is as follows:(13)$$R=10lg\frac{P_{r}}{1\;\mathrm{mW}}$$

Here, R denotes the received signal strength of the UAV to the ground phone during flight, and $P_{r}$ denotes the power of the received signal.

Received Signal Strength Indication (RSSI) is mainly applied to measure the distance between the transmitter and the receiver, and its range measurement method is implemented mainly based on the principle that the power of the radio wave signal decays with the increase in the propagation distance during the transmission of the radio wave in the medium. Therefore, the transmit signal power and the received signal power of a known node are obtained from the transmitter, and then the distance magnitude between the transmitter and the receiver can be calculated by the attenuation model between the signal power and the distance. The relationship between received signal power and distance can be expressed as:(14)$$P_{r}=(P_{t}/d_{a})\times n$$

Here, $P_{r}$ denotes the power of the received signal, $P_{t}$ denotes the power of the transmitted signal, $d_{a}$ denotes the distance between the UAV base station and the ground user, and n denotes the propagation factor of the signal. Substituting Equation (14) into Equation (13), the arithmetic expression of the received signal strength versus distance can be obtained as:(15)$$R=10nlg\left(\frac{P_{t}}{d_{a}}\right)$$

The received signal power A between the UAV base station and the ground user’s cell phone at a distance of 1 m can be expressed as:(16)$$A=10nlgP_{t}$$

Substituting Equation (16) into Equation (15) yields:(17)$$R=A-10nlgd_{a}$$

Since the number of trapped people in the ground scenario is M, the RSSI between the cell phone terminal of user m and the UAV base station can be expressed as:(18)$$R_{m}=A-10nlgd_{am}$$

Here, $R_{m}$ denotes the received signal strength of the UAV to the cell phone terminal of ground user m during flight, and $d_{am}$ denotes the distance size between the UAV base station and user m.

From Equation (18), the distance between the aerial UAV base station u and the ground user m can be expressed as:(19)dam=10^absRm−A/10∗n

Here, $abs\left(R_{m}\right)$ denotes the absolute value of the received signal strength of the UAV to the ground user m cell phone terminal during flight, A denotes the received signal power between the UAV base station and the ground user cell phone at a distance of 1 m, and n denotes the propagation factor of the signal.

Considering that in the actual house collapse environment, the specific number of buried people, and the location of buried people are unknown, random generation is used to construct the ground user model. From Equation (2), the number of ground users is m, where $m\in \left[0,M\right]$, and the 3D position coordinates of each user are $\left(x_{i}^{m},y_{i}^{m},z_{i}^{m}\right)$. The UAV base station will collect the received signal strength R from the ground user’s cell phone in real time during the flight and calculate the distance $d_{a}$ between the ground user and the UAV base station at the same time, and finally, the UAV base station will send the search result $f_{R}$ to the ground control station. The search result $f_{R}$ is as follows:(20)$$f_{R}=\left\{\begin{array}{c} 0,\;\;\;\;\;\;d_{a}>D_{r}\\ 1,\;\;\;\;\;\;d_{a}\leq D_{r} \end{array}\right.$$

Here, $D_{r}$ denotes the maximum search radius of the UAV base station for ground users. When the received signal strength between the ground user’s cell phone and the base station exceeds the set threshold, i.e., the distance between the UAV base station and the ground user’s cell phone terminal is less than the maximum search radius of the UAV, the ground console can precisely locate the location information of the ground user.

## 3. DDQN-SSQN UAV Path Planning Algorithm

This chapter first introduces the knowledge of deep reinforcement learning (DRL) algorithms, and then introduces the design ideas and structure of Double DQN-state splitting Q network (DDQN-SSQN) algorithms in detail.

### 3.1. Deep Reinforcement Learning

Reinforcement learning (RL) is an important branch of artificial intelligence today, and is now widely used in robotics, autonomous driving, computer vision, natural language processing, and other fields. The core of reinforcement learning is oriented to an intelligent body (agent) and puts it into a complex, uncertain environment for interaction. The interaction process between the intelligent body and the environment is shown in Figure 3. The interaction process between agent and environment can be represented by the Markov Decision Process (MDP), and the basic framework of MDP is (S, A, R) [35]. The Agent outputs an action A after acquiring a state S in the environment based on the learned experience. action A, when executed in the environment, outputs the next state of the intelligence and the reward r from that action, and the reward r obtained is superimposed as the intelligence interacts with the environment. The ultimate goal of reinforcement learning is to maximize the value of the reward obtained in long term iterative environmental exploration. Deep Reinforcement Learning (DRL) is a decision control system that combines the ideas of feature extraction and neural networks from Deep Learning (DL) based on reinforcement learning. By inputting multidimensional environmental data information, DRL uses the success and failure experience gained during the exploration process as the basis for updating the decision network and optimizing the network parameters in order to obtain the optimal strategy.

Deep Q-network (DQN) is a Q-learning algorithm based on deep learning. The rewards r obtained by taking action A at each state S are usually stored in Q-learning using the form of a table. Since the learning tasks that the Agent has to face in a practical reinforcement learning task are often continuous, there will be an infinite number of state information, which also leads to the problem of dimensional disaster. DQN combines value function approximation and neural network technology and adopts the target network and experience playback method to train the network, which reduces the requirements for storage space and solves the problem of excessive dimension. The value function approximation can be expressed as:(21)$$Q\left(s,a,\theta \right)\approx Q_{\pi }\left(s,a\right)$$

Here, *s* represents the state S, a represents the action A, θ represents the parameter of the network, and $Q_{\pi }\left(s,a\right)$ represents the actual value function. The target *Q*-value function *y* of the DQN algorithm can be expressed as:(22)$$y^{DQN}=\left\{\begin{array}{c} \;\;\;\;\;\;\;\;\;\;\;\;\;\;\;\;\;\;\;\;\;\;\;r\;\;\;\;\;\;\;\;\;\;\;\;\;\;\;\;\;\;\;\;\;\;\;\;\;,\;end=True\\ r+\gamma \underset{\;a\epsilon A\left(s_{t+1}\right)}{arg\max}Q\left(s_{t+1},\;a;\theta \right),\;end=False \end{array}\right.$$

Here, r represents the reward value obtained and γ represents the discount factor.

Double DQN (DDQN) is an improvement on DQN. The model structure of DDQN and the model structure of DQN are basically the same; the difference lies in their objective function. The objective function of DDQN can be expressed as:(23)$$y^{DDQN}=\left\{\begin{array}{c} \;\;\;\;\;\;\;\;\;\;\;\;\;\;\;\;\;\;\;\;\;\;\;\;\;\;\;\;\;\;\;\;\;r\;\;\;\;\;\;\;\;\;\;\;\;\;\;\;\;\;\;\;\;\;\;\;\;\;\;\;\;\;\;\;\;\;\;\;\;\;\;,\;end=True\\ r+\gamma Q\left(\underset{\;\;\;\;\;\;\;\;\;\;\;\;\;\;a\epsilon A\left(s_{t+1}\right)}{s_{t+1,\;\;\;}arg\max}Q\left(s_{t+1},\;a;\theta \right);\theta^{\prime }\right),\;end=False \end{array}\right.$$

Here, $\theta^{\prime }$ is the periodic copy of θ, which represents the target network parameters. When DDQN calculates the target value, it is calculated not according to the parameters of the target Q network such as DQN, but according to the parameters of the current Q network. Therefore, the calculated target value will be a little smaller than the original. This reduces the problem of overvaluation and makes the *Q* value closer to the true value.

Dueling DQN is also improved on the basis of DQN, and the modified point is at the last layer of the DQN neural network. Originally, the last layer of the DQN neural network was the fully connected layer, and the output after passing through this layer was n *Q* values. Dueling DQN does not directly train to get these n *Q* values, it obtains two indirect variable state values V and act ion advantage A through training, and then expresses the *Q* value through their sum [36]. The calculation formula of the Q function can be expressed as:(24)$$Q\left(s,a;\theta ,\alpha ,\beta \right)=V\left(s;\theta ,\beta \right)+A\left(s,a;\theta ,\alpha \right)$$
where θ is the convolutional layer parameter, and β and α are the two-branch fully connected layer parameters. In practice, action advantage A is generally set as a separate action advantage function minus the average of all action dominance functions in state s [37], The specific formula is as follows:(25)$$Q\left(s,a;\theta ,\alpha ,\beta \right)=V\left(s;\theta ,\beta \right)+(A\left(s,a;\theta ,\alpha \right)-\frac{1}{\left|A\left(s\right)\right|}\underset{a\in A\left(s\right)}{\sum }A\left(s,a;\theta ,\alpha \right)$$

This can narrow the range of *Q* values without affecting the size ordering of the dominant function, which greatly improves the stability of the algorithm.

### 3.2. DDQN-SSQN Algorithm Design

In this paper, an improved algorithm DDQN-SSQN for split-state Q network is proposed based on Double DQN (DDQN) algorithm and Dueling DQN algorithm. The DDQN-SSQN algorithm was used to solve the path decision problem of UAV in flight process. In this algorithm, the process of executing a task by a UAV base station in a simulated scenario was modeled, and the whole model consists of five parts: intelligent body, state, action, reward, and task completion. The modules were built in the following manner.

(1) Agent: The UAV base station will fly as an Agent in the mission area and collect environmental data information in real time. The UAV will select the next action to be completed according to the current state $s_{0}$ and the internal policy π. When the $a_{0}$ action is completed, the UAV will be in a new state $s_{1}$, after which the next action $a_{1}$ will be obtained through the policy π, and so on until the mission is completed.

(2) State: the state $S=\left(p_{o},p_{c},d_{c},d_{a},t_{r},c\right)$ that the UAV is in during flight. Where $p_{o}$ denotes the two-dimensional coordinates of the location of the UAV $(x_{s\_o},y_{s\_o}$), $p_{c}$ denotes the two-dimensional location coordinates of the obstacle detected within the sight range of the UAV $\left(x_{s\_c},y_{s\_c}\right)$, $d_{c}$ denotes the size of the horizontal distance between the UAV and the obstacle, $d_{a}$ denotes the distance between the UAV base station and the trapped person on the ground, $t_{r\_a}$ denotes the degree of convergence of the UAV moving toward the trapped person at the current moment compared to the previous moment. The expression for $t_{r\_a}$ is as follows:(26)$$t_{r\_a}=d_{a}\left(t\right)-d_{a}\left(t-1\right)$$c indicates the number of cell phone locations of people trapped on the ground that were successfully searched by the UAV base station.

(3) Action: Since the flight height of the UAV is fixed in the air 10 m away from the ground, the flight action of the UAV can be regarded as a two-dimensional plane flight problem. Flight action A consists of $\left(a_{1},a_{2},a_{3},a_{4}\right)$. $a_{0}$ means one step forward, $a_{1}$ means one step to the left, $a_{2}$ means one step to the right, and $a_{3}$ means one step backward.

(4) Reward: The real-time reward r obtained by the UAV in the process of completing the mission is determined by the state $s_{t}$ the UAV is in at that moment and the action $a_{t}$ to be performed next, so the reward obtained at moment t can be expressed as $r_{t}\left(s_{t},a_{t}\right)$. The reward $r_{t}\left(s_{t},a_{t}\right)$ obtained by the Agent at each moment consists of five components: the reward $r_{t}^{1}$ obtained by approaching the obstacle, the reward $r_{t}^{2}$ obtained by collision, the reward $r_{t}^{3}$ obtained by approaching the target, the reward $r_{t}^{4}$ obtained by completing the task, and the reward $r_{t}^{5}$ obtained by the loss of raw material of the UAV. $r_{t}\left(s_{t},a_{t}\right)$ can be calculated as follows:(27)$$r_{t}\left(s_{t},a_{t}\right)=\varepsilon_{1}r_{t}^{1}+\varepsilon_{2}r_{t}^{2}+\varepsilon_{3}r_{t}^{3}+\varepsilon_{4}r_{t}^{4}+\varepsilon_{5}r_{t}^{5}$$

Here, $\varepsilon_{1}$, $\varepsilon_{2}$, $\varepsilon_{3}$, $\varepsilon_{4}$ and $\varepsilon_{5}$ denote the weights of the rewards obtained by completing the corresponding target tasks, respectively, and the sum of all reward weights satisfies:(28)$$\varepsilon_{1}+\varepsilon_{2}+\varepsilon_{3}+\varepsilon_{4}+\varepsilon_{5}=1$$

The UAV can detect the horizontal distance $d_{c}$ between the airframe and the surrounding obstacles in real time by the millimeter wave radar it carries. Since the detection range of the millimeter wave radar is limited and the threat of the obstacle to the airframe itself is minimal when the horizontal distance between the obstacle and the UAV is more than 10 m, $d_{c}\in \left(0,10\right]$ is set. Therefore, the proximity to the obstacle reward $r_{t}^{1}$ can be expressed as:(29)$$r_{t}^{1}=\left\{\begin{array}{c} \mu_{1}*\left(d_{c}\left(t\right)-d_{c}\left(t-1\right)\right),d_{c}\in \left(0,10\right]\\ \;\;\;\;\;\;\;\;\;\;\;\;\;\;\;\;\;\;\;\;0\;\;\;\;\;\;\;\;\;\;\;\;\;\;\;\;\;\;,\;d_{c}>10 \end{array}\right.$$

Here, $\mu_{1}$ denotes the reward factor for proximity to an obstacle.

Collision reward is the reward obtained when a UAV collides with an obstacle in its flight path. However, according to the actual knowledge, the UAV base station is costly and easily damaged, and once it collides with an obstacle it will cause huge property damage. Therefore, the collision range is set to 2 m in the collision model, i.e., a collision event is considered to occur when the distance $d_{c}$ between the UAV and the obstacle is less than 2 m. Then the collision reward $r_{t}^{2}$ can be expressed as:(30)$$r_{t}^{2}=\left\{\begin{array}{c} 100*\mu_{2}\;,\;\;d_{c}\leq 2\\ \;\;\;\;\;\;\;\;\;\;\;\;\;\;0\;\;\;\;\;\;,\;\;d_{c}\epsilon \left(2,10\right] \end{array}\right.$$

The RSSI signal data between the airborne base station and the ground personnel’s cell phone are continuously collected by the UAV base station in the course of the disaster relief mission, and are used to obtain the horizontal distance $d_{c}$ of the affected personnel from the UAV and the convergence degree $t_{r\_a}$ of the UAV to the location of the affected personnel at the current moment compared to the previous moment. Therefore, the proximity to the target reward $r_{t}^{3}$ can be expressed as:(31)$$\;r_{t}^{3}=\mu_{3}*\left(t_{r_{a}}+\frac{c}{10}\right)$$

Here, $\mu_{3}$ represents the reward factor close to the target, and c represents the number of people trapped on the ground that the UAV searches.

The UAV base station establishes information communication with the trapped person’s cell phone to obtain the received signal strength indication RSSI and determines the direction and specific location of the trapped person’s cell phone according to the change of signal strength. When the distance between the UAV and the ground user’s mobile phone terminal is long due to low communication signal strength, large signal interference, and other reasons, it will be impossible to accurately locate the location of the trapped person, so when the horizontal distance between the UAV base station and the ground mobile phone terminal is less than 5 m, it can be considered that the disaster victim has been successfully found. If the location of all trapped persons is found, the mission is considered complete. Task completion reward $r_{t}^{4}$ can be expressed as:(32)$$\;r_{t}^{4}=100*\mu_{4}$$

Here, $\mu_{4}$ represents the reward coefficient for task completion.

In the process of performing tasks, drones will inevitably have problems with power consumption and hardware equipment consumption. UAV raw material loss reward $r_{t}^{5}$ can be expressed as:(33)$$r_{t}^{5}=10*\mu_{5}$$

Here, $\mu_{5}$ represents the reward coefficient of the loss of raw materials for the UAV.

(5) Task completion: The completion of UAV flight tasks $f_{d}$ determined by the maximum number of flight steps *p* and the number of UAV base stations successfully searched for ground mobile phone terminals *c*. The expression for task completion $f_{d}$ is as follows:(34)$$f_{d}=\left\{\begin{array}{c} \;\;\;\;0,\;\;\;p<P_{MAX}\;and\;c<M\\ \;\;\;1,\;\;\;p=P_{MAX}\;\;or\;\;c=M \end{array}\right.$$

Among them, when $f_{d}=0$, it means that the search and rescue mission is still in progress; When $f_{d}=1$, it means that the UAV has completed this search and rescue mission, the $P_{MAX}$ indicates the maximum number of flight steps of the UAV, and M represents the number of trapped people on the ground.

### 3.3. DDQN-SSQN Algorithm Structure

The DDQN-SSQN algorithm proposed in this paper is an improvement on the neural network based on the traditional DDQN algorithm. The network structure of the DDQN-SSQN algorithm is shown in Figure 4, where $\left(S_{1}\sim S_{8}\right)$ denotes the state information of the intelligent body, mainly including the position coordinates $p_{o}\left(x_{s\_n},y_{s\_n}\right)$ of the UAV, the position coordinates $p_{c}\left(x_{s\_c},y_{s\_c}\right)$ of the obstacle, the horizontal distance $d_{c}$ between the obstacle and the UAV, the straight-line distance $d_{a}$ between the trapped person and the UAV, the convergence degree $t_{r\_a}$ of the UAV moving toward the trapped person, and the number of people c who searched for the trapped person. Before the state information of the drone is input into the neural network, the overall state State is divided into three parts: position state State1 $\left(S_{1}\sim S_{4}\right)$, distance state State2 $\left(S_{5},S_{6}\right)$ and other state State3 $\left(S_{7},S_{8}\right)$. State1 $\left(S_{1}\sim S_{4}\right)$ is predicted by hidden layer 1 and hidden layer 2 to obtain the position state of UAV and obstacle at the next moment ($S_{1}^{\prime }\sim S_{4}^{\prime }$); State2 $\left(S_{5},S_{6}\right)$ and ($S_{1}^{\prime }\sim S_{4}^{\prime }$) are predicted by hidden layer 3 and hidden layer 4 to obtain the position state and distance state at the next moment ($S_{1}^{\prime \prime }\sim S_{4}^{\prime \prime },S_{5}^{\prime },S_{6}^{\prime }$); State3 $\left(S_{7},S_{8}\right)$ and ($S_{1}^{\prime \prime }\sim S_{4}^{\prime \prime },S_{5}^{\prime },S_{6}^{\prime }$) output action value $A_{i}\left(s,a\right)$ and state advantage $V\left(s\right)$ through hidden layer 5 and hidden layer 6, and then sum up the two to finally obtain $Q_{i}\left(s,a\right)$. $Q_{i}\left(s,a\right)$ represents the value of each action performed by the Agent in the current state and is calculated as:(35)$$Q_{i}\left(s,a\right)=\left(A_{i}\left(s,a\right)-\frac{1}{\left|A\left(s\right)\right|}\underset{a\in A\left(s\right)}{\sum }A\left(s,a\right)\right)+V\left(s\right)$$
Here, $i\in \left[1,4\right]$, *s* represents the state S, a represents the action A. 

The execution architecture of the DDQN-SSQN algorithm is shown in Figure 5. The action policy used by the algorithm is the ε-greedy policy [38], i.e., the Agent has a probability of executing the action corresponding to the maximum Q function in the scene model with 1−ε. The execution action *A* can be expressed as:(36)$$A=\left\{\begin{array}{c} arg\underset{a}{\max}Q\left(s,a\right),\;\;P<\left(1-\varepsilon \right)\\ Random\;\;\;\;\;\;\;\;,\;\;P\geq \left(1-\varepsilon \right) \end{array}\right.$$

Here, $argmaxQ\left(s,a\right)$ denotes the action corresponding to the maximum value in the $Q\left(s,a\right)$ function.

The DDQN-SSQN algorithm proposed in this paper is summarized in Algorithm 1, where the input is composed of the UAV’s own flight parameters, agent model, and environmental parameters (step 1), and the output is the set of relevant data parameters obtained during training and the trained agent model (step 29).

In the model training phase, relevant training parameters such as the number of training rounds, discount factor, experience pool capacity, and update frequency are first initialized. Next, set the number of obstacles and trapped people, and clear the target Q network of the agent (step 2).


**Algorithm 1:** The Proposed DDQN-SSQN Algorithm Scheme

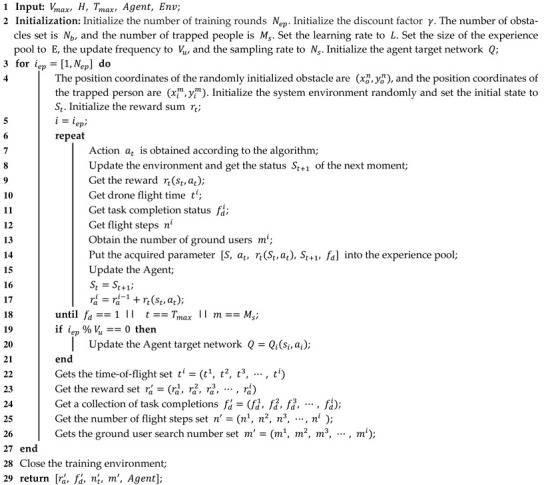



At the beginning of each round of training, the position coordinates of the obstacles and the position coordinates of the trapped people are randomly reset; the mission environment faced by the drone will reset and obtain the initial state $S_{t}$; even if the reward received by the agent is cleared to zero (step 4). The training of the model will undergo $N_{ep}$ iterations (step 3).

In the process of training, the agent will obtain the next action $a_{t}$ to be performed according to the current state $S_{t}$ (step 7), and the selection of actions will be completed using the ε-greedy strategy (Equation (32)); After the agent executes action $a_{t}$, the task environment will be updated, and a new status $S_{t+1}$ will be obtained, and the reward $r_{t}\left(s_{t},a_{t}\right)$ obtained for performing the action will be obtained (steps 8–9); During the execution of training, the agent will collect real-time task parameter information, including: task completion, flight step, flight time, and the number of trapped people found (steps 10–13); The collected data information is packaged and sent to the experience pool (step 14) to provide a data set for the agent’s learning. After the above data collection is completed, the system will update the status information $S_{t}$ and the agent in time, and obtain the sum of the rewards $r_{a}^{i}$ obtained by the agent from the beginning of the first round to the end of the i round (steps 15–17). During each turn, the agent repeats the flight mission in the simulation environment until the flight mission is completed, the flight time exceeds the set threshold, or all the difficult personnel are found, and the flight mission is terminated (steps 6–18). After the task is terminated, the various data obtained up to this turn will be packaged and stored in the form of a collection (steps 22–26), and if the number of training rounds meets the system update frequency, the target Q network will be updated (steps 19–21). After this round of training, a new round of training will be restarted, and training will not stop until the number of training rounds reaches $N_{ep}$. Finally, the parameter information of various tasks obtained during training will be output, and a trained agent model will also be obtained. After that, the drone will use the trained model to carry out real-time path planning in the disaster relief environment and the location information collection of trapped people.

## 4. Experimental Simulation and Analysis

This chapter focuses on the experimental simulation design. First, the algorithm is simulated and analyzed using the Open AI Gym simulation platform, and then the DDQN-SSQN algorithm is compared with the traditional DDQN, DQN, and Q-Learning algorithm schemes to verify the superiority of the algorithm proposed in this paper.

### 4.1. Simulate Experimental Design

The simulation parameters of the scene where the UAV base station collects the location information of the affected personnel were as follows: the flight area of the UAV base station was set to 20 × 20, and the entire area was composed of 400 rectangles with a side length of 5 m, so the task range of the scene was 100 m × 100 m. The flight height of the UAV was fixed in the air at a height of H=10 m from the ground, the maximum flight speed was set to $V_{max}=5\;m/s$, and the maximum flight time was set to $T_{max}=140\;\min$. The obstacles encountered by the drone during flight randomly appeared in the scene model, the number of generated obstacles in the scene model $N_{b}=20$, and the two-dimensional position coordinates of the obstacles were $\left(x_{o}^{n},y_{o}^{n}\right)$, of which $x_{o}^{n}\epsilon \left[0,19\right]$, $y_{o}^{n}\epsilon \left[0,19\right]$, $n\epsilon \left[1,20\right]$. In the scene model, the ground disaster victims appeared randomly, the number of disaster victims generated in the model was $M_{s}=500$, and the two-dimensional location coordinates of the ground users were $\left(x_{i}^{m},y_{i}^{m}\right)$, where $x_{i}^{m}\epsilon \left[0,19\right]$, $y_{i}^{m}\epsilon \left[0,19\right]$, $m\epsilon \left[1,500\right]$. Table 1 describes the simulation parameters during training.

### 4.2. Analysis of Simulation Result

This paper used the PyTorch deep learning architecture to complete the network part of the deep reinforcement learning algorithm. When the drone collided with an obstacle or the flight time of the UAV was greater than $T_{max}$, the training round was set to end. The training reward results of the four algorithms in the simulation scenario are shown in Figure 6.

According to Figure 6, in the first 100 rounds of training, the four algorithms were in the environmental exploration stage, and the reward value obtained by the agent oscillated disorderly; In 100~200 rounds, the rewards obtained by the DDQN-SSQN algorithm began to rise rapidly, the rewards obtained by the DDQN and DQN algorithms rose slowly, and the rewards obtained by the Q-Learning algorithm did not see an upward trend; When the number of training rounds reached 200 rounds, the reward return of the DDQN-SSQN algorithm began to converge and eventually stabilized, while the remaining three algorithms began to converge after the training round reached 400 rounds, and were not completely stable before 500 rounds and still had a large oscillation amplitude. The DDQN-SSQN algorithm proposed in this paper can accelerate the learning process by learning the environmental state, and compared with the other three algorithms, the DDQN-SSQN algorithm can provide the best flight strategy for the UAV in a shorter time and obtain more rewards.

In Figure 7a–d show the obstacle avoidance path diagram obtained by UAV training under the four algorithms of DQN, DDQN, Q-Learning, and DDQN-SSQN, respectively. By observing the above path diagram, it can be seen that the drone can effectively avoid obstacles under the training of the four algorithms, but the path taken by the drone was different. In the obstacle avoidance path diagram of the Q-Learning algorithm shown in Figure 7c, the drone consumed the most steps, and the search strategy selected by the algorithm was to fly along the edge of the obstacle, although this method can avoid obstacles, but also increased the risk of drone crash; Figure 7a shows the obstacle avoidance path of the DQN algorithm, and it can be seen that the UAV calculation strategy was to search away from obstacles, so that although the threat of obstacles can be effectively avoided, the UAV’s search coverage of the mission area was reduced; Figure 7b shows the obstacle avoidance path of the DDQN algorithm, which was significantly better than that of Figure 7a and Figure 7c, but repeated paths appeared during the UAV search process, which increased the energy consumption of the UAV; It can be seen from the UAV obstacle avoidance path diagram under the DDQN-SSQN algorithm shown in Figure 7d that under this scheme, the UAV could complete the entire search task with the least number of flight steps under the condition of ensuring the coverage of the mission area, effectively reducing the flight energy consumption of the UAV. Therefore, the DDQN-SSQN algorithm scheme was better than other schemes.

The time taken by the UAV to complete the flight task under the control of four algorithms was compared, and the comparison results are shown in Figure 8.

According to Figure 8, in the first 250 rounds, because the UAV needed to explore the environment and gain experience, it took more time for the four algorithms to complete a task cycle in the early stage; When the training reached 250~300 rounds, the time required for the search and rescue UAV based on the DDQN-SSQN algorithm to complete the task was greatly reduced, and the time consumed after 300 rounds began to stabilize, about 35~38 min. From the perspective of the overall convergence trend, the Q-Learning algorithm scheme took the longest time to complete the task, about 50~60 min, while the time consumed by DDQN and DQN algorithm schemes decreased after 450 rounds, but it was always higher than the algorithm scheme proposed in this paper. Therefore, the DDQN-SSQN algorithm scheme was superior to other schemes, under which the UAV could complete the search and rescue task in the shortest time.

Figure 9 shows the 3D path planning diagram of the UAV under the four algorithms of DQN, DDQN, Q-Learning, and DDQN-SSQN. The UAV made an optimal path planning solution based on the size of the RSSI value and the coordinates of the obstacle’s location. Figure 10 shows the comparison graph of the number of people searched by UAVs under the four algorithms, from which it can be seen that the number of people searched by the three algorithms DQN, DDQNm and DDQN-SSQN in rounds 0~100 rose rapidly and the Q-Learning algorithm rose very slowly; When the training reached 100~200 rounds, the rising trend of the DQN and DDQN algorithms moderated and oscillated a little, and the DDQN-SSQN algorithm continued to rise rapidly and started to stabilize after 200 rounds; After 250 rounds the DDQN algorithm started to stabilize gradually, while the DQN, Q-Learning algorithm remained in oscillation until the end of training. From the above analysis, it can be seen that the proposed scheme could search for more affected people in the shortest time, so the DDQN-SSQN algorithm scheme was better than other schemes.

The number of buried people found throughout the search task was used as an indicator of task completion. The more buried people found in a task cycle, then the higher the task completion rate. Figure 11 shows the completion of the search and rescue tasks by the UAV base station under the manipulation of the four algorithms. From the figure, it can be seen that the task completion rate of all four algorithm schemes gradually increased with the increase in training rounds, among which the DDQN-SSQN algorithm scheme had the fastest increase in task completion rate and ccould complete the whole search and rescue task after the training reached 250 rounds. Figure 12 shows a comparison chart of the success rate of UAV obstacle avoidance under four algorithm schemes. The four stages in the figure represent the number of rounds in the training process, 1, 2, 3, 4 correspond to 0~200 rounds, 200~400 rounds, 400~600 rounds, and 600~700 rounds, respectively. It can be seen from the figure that the success rate of DDQN-SSQN algorithm scheme in obstacle avoidance was much higher than that of the remaining three algorithms during the training process. According to the above analysis, the DDQN-SSQN algorithm enabled the UAV to maximize the completion rate of search and rescue tasks with the shortest number of training rounds without collision. Therefore, the proposed algorithm scheme was better than the other three algorithms.

## 5. Conclusions

This paper investigated the path planning problem in the process of collecting the location information of the affected people by UAV base station based on deep reinforcement learning algorithm and proposed a DDQN-SSQN reinforcement learning algorithm combining state splitting and optimal state to complete the optimal path planning in the process of UAV information collection. In order to realize the collection of the location information of the trapped people on the ground by the UAV base station, we introduced RSSI as the search indication of the disaster victims and input tedit into the algorithm network as status information for learning, which reduced the difficulty of search and rescue decision-making of the UAV base station. In this paper, the virtual disaster relief scenario model was constructed using the Open AI Gym simulation platform, and the network structure of the algorithm was improved on the basis of the DDQN algorithm, and then the new algorithm designed in this paper was compared with the traditional reinforcement learning algorithm through simulation comparison experiments. The simulation experimental results showed that the proposed algorithm could find the optimal search path in the shortest time compared with several other reinforcement learning algorithms. The proposed algorithm was better than the traditional algorithm in many aspects such as training speed, task completion rate, and obstacle avoidance rate. In future research, we plan to consider the constraints related to terrain and environment when dividing the mission area, so as to avoid wasting resources due to UAV search and exploration in uninhabited land. In addition, the location information of the trapped people on the ground collected by the UAV can be further processed to achieve three-dimensional coordinate positioning of the disaster-stricken people.

## Figures and Tables

**Figure 1 entropy-24-01767-f001:**
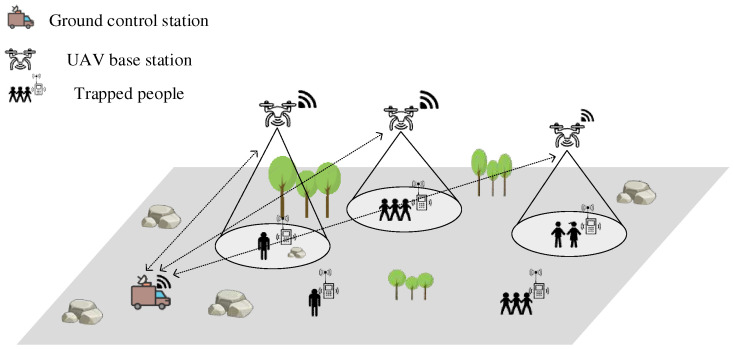
System environment model.

**Figure 2 entropy-24-01767-f002:**
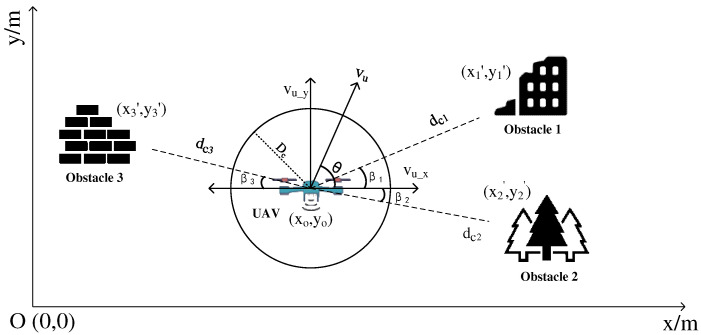
UAV obstacle avoidance detection.

**Figure 3 entropy-24-01767-f003:**
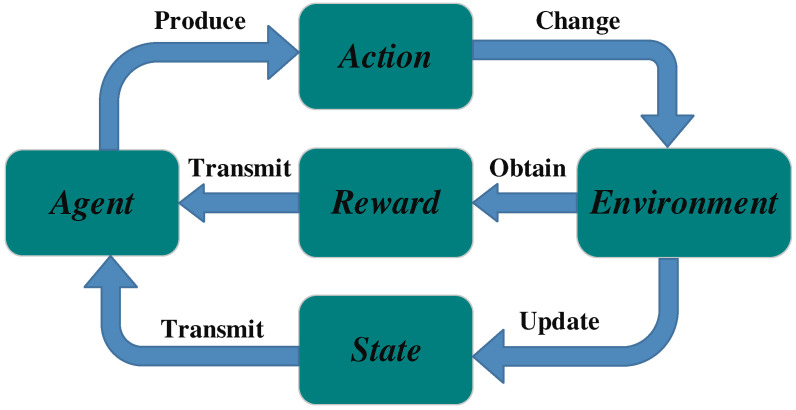
Reinforcement learning schematic.

**Figure 4 entropy-24-01767-f004:**
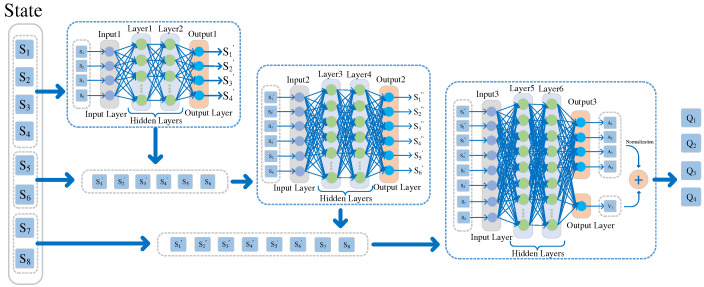
Network structure of DDQN-SSQN algorithm.

**Figure 5 entropy-24-01767-f005:**
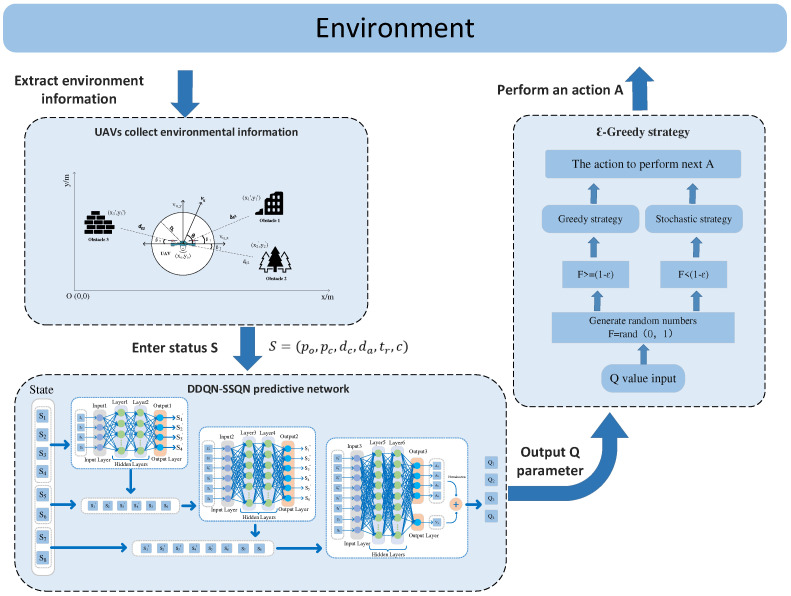
DDQN-SSQN algorithm execution architecture.

**Figure 6 entropy-24-01767-f006:**
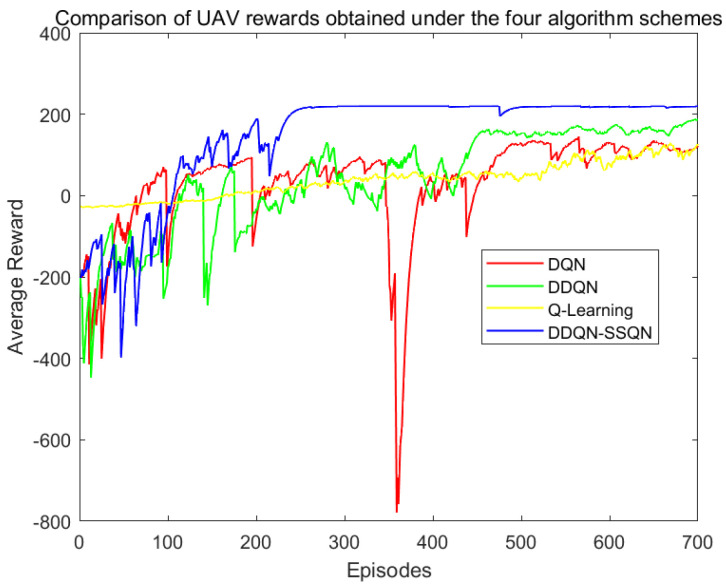
Comparison chart of rewards earned.

**Figure 7 entropy-24-01767-f007:**
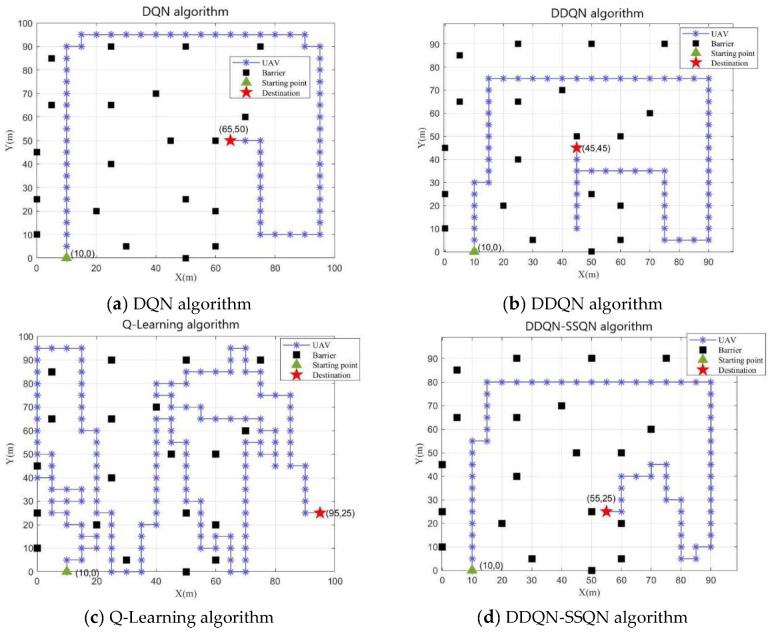
UAV obstacle avoidance path diagram under four algorithm schemes.

**Figure 8 entropy-24-01767-f008:**
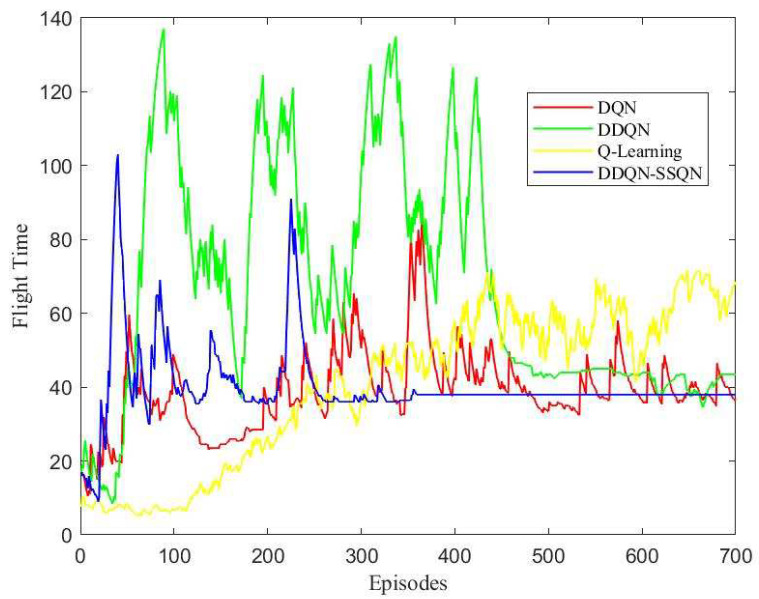
Comparison of UAV flight time under four algorithm schemes.

**Figure 9 entropy-24-01767-f009:**
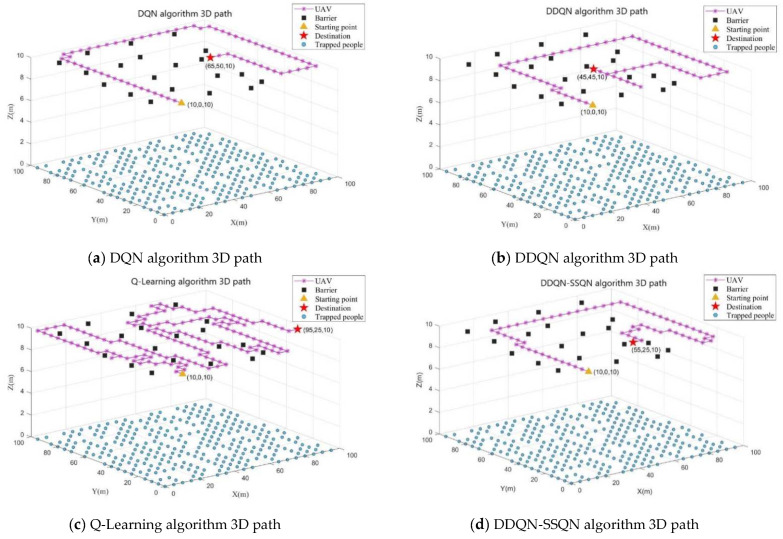
Three-dimensional path planning of UAV under four algorithm schemes.

**Figure 10 entropy-24-01767-f010:**
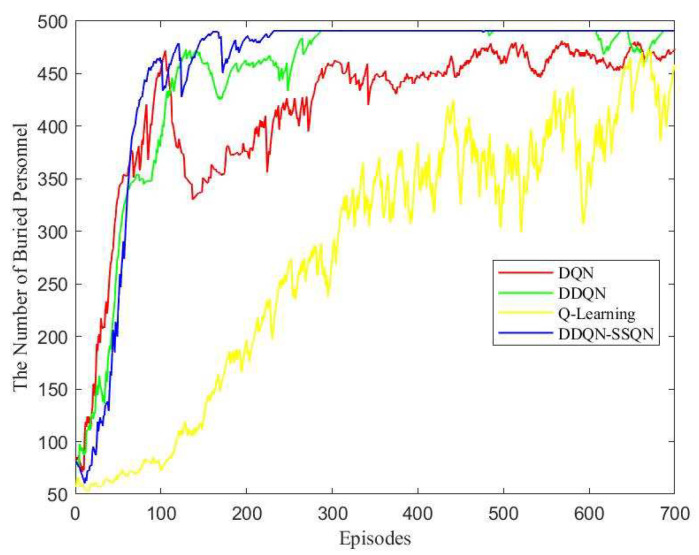
Comparison of the number of UAVs searching for disaster victims under four algorithm schemes.

**Figure 11 entropy-24-01767-f011:**
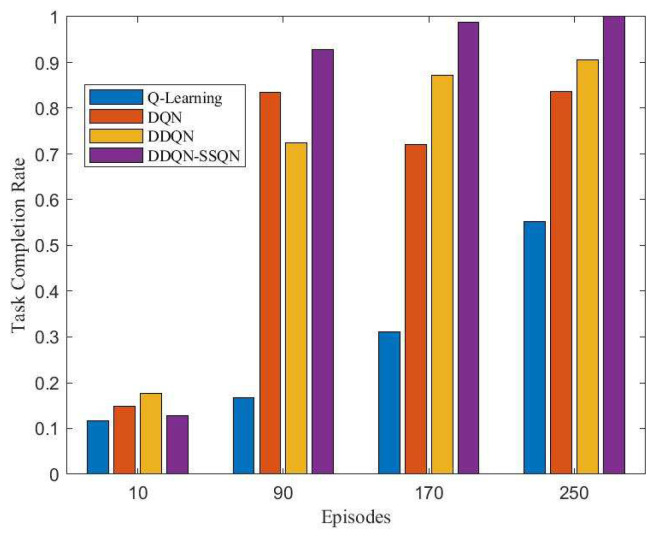
Comparison of UAV mission completion rates under four algorithm schemes.

**Figure 12 entropy-24-01767-f012:**
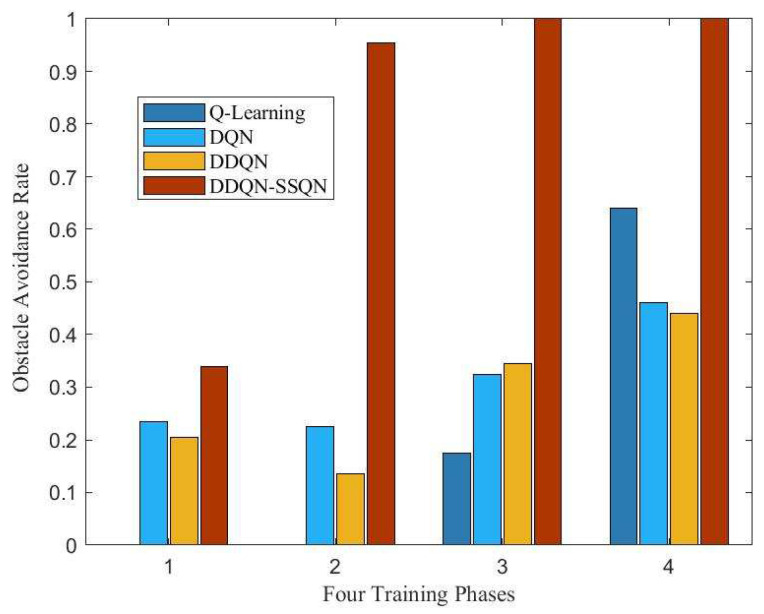
Comparison of the success rate of UAV obstacle avoidance under four algorithm schemes.

**Table 1 entropy-24-01767-t001:** Training simulation parameters.

Parameter Symbols.	Parameter Name	Parameter Setting Value
$S_{n}$	Number of states	8
$A_{n}$	Number of movements	4
$N_{ep}$	Number of training rounds	700
γ	Discount Factor	0.95
L	Learning Rate	0.0001
E	Experience pool capacity	100,000
$V_{u}$	Update Frequency	2
$N_{s}$	Number of samples taken	128

## Data Availability

Not applicable.

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
