# Peer review of "Path Planning Research of a UAV Base Station Searching for Disaster Victims’ Location Information Based on Deep Reinforcement Learning"

_entropy, 2022, doi:10.3390/e24121767_

Round 1

Reviewer 1 Report

Section

Comment

Suggestion

Abstract

Line 9-10

For the path planning problem of unmanned aerial vehicle (UAV) base stations in the 9 process of searching the location information of trapped people on the ground, this paper proposes

The sentence is not clear – maybe the English has to be reviewed

Modify : re-phrase the sentence

Abstract

Line 11

 Acronyms used in the Abstract should be spelled out

Modify: Clarify the acronyms

Section 1

Line 27

flexible airframe deployment

The UAV base station, i.e., the UAV ground station is in charge of flexible airframe deployment ?

Sometimes maybe. In any case the expression airframe deployment is unusual- what is it meaning? Airframe is intended as the structure of the UAV without equipment – do you mean this? 

Modify – clarify the concepts

Line 129 to Line 133

The period is repeated twice

Modify - Remove the repetition

Line 141 -144

The sentences refer to the following terms tasks, mission subtasks.

They should be clarified before , otherwise the definition may seem not rigorous.

If there are N drones and N area the areas – it may seem that the problem is to cover N area with N drones independently of the topography.

The concept needs to be described better

Modify - Explain better

Line 148

The size of the ? ′ sub-task area ??? ? obtained from the scene model segmentation  will be set according to the performance of ? UAVs

If we look at the setting of the problem inside out. It seems that the paper is saying. N drones with they performance may cover N areas – which extension depends on drone performance. Hence,  it is not said that the area covered is equal to the Area of interest .

But at line 141 it is said that the Area of Interest is dived in N’ parts that have to be equal to the number of drones and if they cannot cover the extension?

It has to be described clearly the constraint of the problem to be solved.

Formula at line 7 is clear but highlights the inconsistency with the arbitrariness of N’ driven by the topography as declared at line 143

Modify – clarify better

Section 2.2 Line 156-163

The overall period is clear. It is also true that the issue of the obstacles depends on the altitude of the flight, that on turn depends on the type of sensors the UAV are equipped.

Accordingly, instead of generic sentences saying that after a disaster the environment is complex, the paper should better outline the problem and the constraints in the introduction.

To be clear to most of the readers the paper should state

Modify the introduction

Line 341

If the location of all trapped persons is found, the mission is 391 considered complete

It represents a constraint that is never mentioned before in the formalization of the problem.

Is a number that is known a priori when a UAV is assigned to a certain area?

Modify adding a clarification on this point

Conclusion

Future works seems to be too short. For example what happens if the number of UAV is less than the number N’?

Add further consideration

Reviewer 2 Report

Check spacing between Fig. and the numbers.

Introduce Figures in the text before showing them. Seeing a picture and not knowing why it is there can become confusing.

On page 12, "-greedy greedy strategy" - is this correct? Epsilon missing? Is the double "greedy" next to each other needed?

Don't have headings next to each other. E.g. heading 4 and heading 4.1. Separate them with a sentence or two elaborating the sub-sections to follow.

Insert a space between values and units. E.g. "5 m" instead of "5m".

It is stated that the DDQN-SSQN algorithm is better. But what flaws were encountered? What improvements can be made?

Buried people are mentioned, but the algorithm is related to obstacle avoidance. It seems to go off topic of the title?

What re the contributions of the paper? These should be listed in bullet points in the last paragraph of the introduction. They should be given again in the conclusion, but indicate how they were achieved in the paper.

Reviewer 3 Report

In this paper, authors proposed an algorithm, named DDQN-SSQN to solve the path planning problem. By combining DDQN and SSQN, authors obtained the optimal path faster than other traditional algorithmic schemes. It is interesting that the stability and convergence speed of the algorithm were quite good. I have some following comments.

a.      It is better to mention the term SSQN in the introduction section.

b.      It is wonderful that the stability and convergence speed of the algorithm are good. But I wonder about the complexity of the proposed algorithm. Have you considered some factors when implementing such an algorithm in practical scenario.

c.      I agree with the authors that the proposed scheme can plan the optimal path faster than other traditional algorithmic schemes. Could you please discuss further about the speediness of the algorithm?

d.      What is the Epsilon_5 in the Eq. (27) ?
